# Predictors of Length of Hospitalization and Impact on Early Readmission for Mental Disorders

**DOI:** 10.3390/ijerph192215127

**Published:** 2022-11-16

**Authors:** Lia Gentil, Guy Grenier, Helen-Maria Vasiliadis, Marie-Josée Fleury

**Affiliations:** 1Department of Psychiatry, McGill University, 1033, Pine Avenue West, Montreal, QC H3A 1A1, Canada; 2Douglas Hospital Research Centre, Douglas Mental Health University Institute, 6875 LaSalle Blvd, Montreal, QC H4H 1R3, Canada; 3Département Des Sciences de la Santé Communautaire, Université de Sherbrooke, Longueuil, QC J4K 0A8, Canada; 4Centre de Recherche Charles-Le Moyne-Saguenay-Lac-Saint-Jean sur les Innovations en Santé (CR-CSIS), Campus de Longueuil-Université de Sherbrooke, 150 Place Charles-Lemoyne, Longueuil, QC J4K 0A8, Canada

**Keywords:** length of hospitalization, early readmission, emergency departments, mental disorders, predictors, brief-stay hospitalization, long-stay hospitalization

## Abstract

Length of hospitalization, if inappropriate to patient needs, may be associated with early readmission, reflecting sub-optimal hospital treatment, and translating difficulties to access outpatient care after discharge. This study identified predictors of brief-stay (1–6 days), mid-stay (7–30 days) or long-stay (≥31 days) hospitalization, and evaluated how lengths of hospital stay impacted on early readmission (within 30 days) among 3729 patients with mental disorders (MD) or substance-related disorders (SRD). This five-year cohort study used medical administrative databases and multinomial logistic regression. Compared to patients with brief-stay or mid-stay hospitalization, more long-stay patients were 65+ years old, had serious MD, and had a usual psychiatrist rather than a general practitioner (GP). Predictors of early readmission were brief-stay hospitalization, residence in more materially deprived areas, more diagnoses of MD/SRD or chronic physical illnesses, and having a usual psychiatrist with or without a GP. Patients with long-stay hospitalization (≥31 days) and early readmission had more complex conditions, especially more co-occurring chronic physical illnesses, and more serious MD, while they tended to have a usual psychiatrist with or without a GP. For patients with more complex conditions, programs such as assertive community treatment, intensive case management or home treatment would be advisable, particularly for those living in materially deprived areas.

## 1. Introduction

Most individuals with mental disorders (MD), including substance-related disorders (SRD), are treated in outpatient care, and hospitalized relatively infrequently [1]. As hospitalization is one of the costliest healthcare services, current trends favoring reduced lengths of stay and brief hospitalization appear justified [2]. However, length of hospital stay varies greatly depending on patient characteristics, access to services, and clinical practices. Both long-stay [3] and brief-stay hospitalizations [4,5] have been associated with early readmission (within 30 days of discharge) [6], limiting recommendations related to best practices. Rates of early readmission following hospital discharge, identified in psychiatry as a key adverse outcome [7,8,9], are elevated. Previous studies reported that 8–17% of patients with MD were readmitted early following hospital discharge [10,11,12]. Inadequate length of hospitalization or early readmission may reflect sub-optimal hospital treatment and may further translate into difficulties accessing outpatient care after discharge [6]. Moreover, outpatient care is not always planned effectively at hospital discharge [13]. Early readmission may also undermine patient and clinician confidence in treatment, suggesting negative effects for patient recovery [14].

Some studies have previously assessed associations between length of hospital stay and early readmission, but have mostly considered patient sociodemographic and clinical characteristics. Longer-stay hospitalization and early readmission were both linked with unemployment [2,6,15], homelessness [6,15], having never been married [6,10,16], serious MD [6,17,18], previous hospitalizations [1,19], and lack of discharge planning [1,20]. Longer-stay hospitalizations only were linked with co-occurring MD/physical illnesses [21] and more structured or supervised aftercare [18], while short-stay hospitalizations were associated with patients having more SRD [1,16,22]. Early readmission only was linked with personality disorders [5,10,14,23] and co-occurring MD/physical illnesses [11,24], whereas brief care management [20] and more intensive follow-up care [25] protected against early readmission.

To our knowledge, previous studies have not evaluated predictors of length of hospital stay, comparing brief-stay, mid-stay, or long-stay hospitalization and their respective impacts on early readmission. Associations between patient service use, length of hospital stay, and early readmission are also understudied. This study hypothesized that patients with more complex medical conditions would experience longer-stay hospitalizations and higher rates of early readmission, while more intensive and continuous outpatient care would lower the risks of both adverse outcomes. Whether or not longer-stay hospitalization increased the risk of early readmission was also tested. A better knowledge of predictors for different lengths of hospital stays, and how the length of hospital stays impacts on early readmission may be useful for clinicians and policy makers responsible for the improvement of patient care. This study compared predictors of brief-stay (1–6 days), mid-stay (7–30 days), and long-stay (≥31 days) hospitalization, and evaluated how different lengths of hospital stay impacted on early readmission among patients with MD.

## 2. Materials and Methods

### 2.1. Study Sample, Sources, and Design

This 5-year study was based on medical administrative databases investigating a cohort of 12,000 patients, who were recruited and used in 2014-15 (1 April–31 March) at one of six Quebec (Canada) hospital emergency departments (ED) located in urban areas. Quebec has a universal public healthcare insurance system that covers all physician care and some psychosocial services, mainly for mental health individual or group therapy provided in community healthcare centers. Study patients were required to be 12+ years old, eligible under the Quebec Health Insurance Regime (*Régie d’assurance maladie du Québec*: RAMQ) and hospitalized after their recruitment from 2014-15 to 2016-17 where “index hospitalizations” (first hospitalization in this 3-year following period) for mental health (MH) reasons (MD, SRD, or suicide attempt) were assessed. At least a prior 2-year data period was also needed to identify clinical patient characteristics before patient index hospitalization, measured from 2012-13 up to the index hospitalization. Patient data were also required to be available for at least 30 days following discharge from index hospitalization in order to measure early readmission.

Data extracted from RAMQ included several sub-databases containing information on patient sociodemographic and health characteristics, and health service use throughout Quebec, including hospitalization, ED use, physician care, and psychosocial interventions in community healthcare centers. Only 6% of physicians in Quebec practice outside the public health system (RAMQ) [26]. Data from the sub-databases were merged for each patient using a unique encrypted identifier. Predictors for the two dependent variables, length of hospital stay, and early readmission were grouped according to sociodemographic, clinical, and service use characteristics. Sociodemographic patient characteristics were measured for both dependent variables at index hospitalization from 2014-15 to 2016-17, while clinical patient characteristics were measured from 2012-13 to index hospitalization. Service use predictors for lengths of hospital stay were measured over the 12-month period prior to index hospitalization, while the control variables for early readmission covered the 30-day period following hospital discharge, except for usual physician, which was measured for the 12 months prior to the 30-day period following hospital discharge when early readmission could occur. As the study used a medical administrative database (RAMQ), informed consent from patients was not required. Access to the RAMQ was granted by the Quebec Commission for Access to Information. The ethics committee of a psychiatric hospital approved the study protocol.

### 2.2. Study Variables

The dependent variable “length of hospital stay” included three categories: brief-stay (1–6 days), mid-stay (7–30 days), and long-stay (≥31 days). These were based on an adequate distribution of the cohort’s length of hospitalization (Figure 1), and on a previous study [27]. There are evidence that long-stay hospitalizations (≥31 days) increase the risk of complications or adverse outcomes, especially for patients with chronic physical illnesses [28,29]. Early readmission within 30-days after hospital discharge is a standard indicator of adverse outcome, pointing to inadequate discharge planning or post-discharge follow-up in the community [8,9].

Patient sociodemographic characteristics included sex, age groups, and material and social deprivation. Based on the smallest geographical units established for the 2011 Canadian census, the Material Deprivation Index was used, representing the ratio of population employment, mean income and number of individuals without a high school diploma, while the Social Deprivation Index included individuals living alone, those without a spouse, and single-parent families [30]. These were classified in quintiles, the fifth one representing the highest level of deprivation, but were regrouped for this study into less deprived areas (1, 2, 3) and more deprived areas (4, 5, and “not assigned areas”—e.g., homeless individuals).

Clinical characteristics included MD, SRD, chronic physical illnesses, co-occurring disorders (MD, SRD, chronic physical illnesses), number of MD/SRD diagnoses, and suicidal behaviors. MD included common MD (adjustment, anxiety and depressive disorders, attention deficit/hyperactivity disorder), serious MD (bipolar, schizophrenia spectrum and other psychotic disorders), and personality disorders. SRD were alcohol- or drug-related disorders (use and induced disorders, intoxication, withdrawal). Chronic physical illnesses were adapted from an integrated version of both Elixhauser and Charlson Comorbidity Indexes, including number and severity (0–3+) [31]. RAMQ diagnostic codes were based on the International Classification of Diseases Ninth or Tenth Revision (Appendix A). Suicide ideations or attempts were reported as reasons for index hospitalization or ED use by triage nurses well-trained in the identification of suicidal behaviors, which provided reliable data [32].

Service use variables related to length of hospitalization included: having a usual physician (usual general practitioner [GP] only, psychiatrist only, both GP and psychiatrist), number of consultations with usual GP or psychiatrist, high continuity of physician care, number of psychosocial interventions in community healthcare centers, and previous hospitalization or high ED use for MH reasons prior to the 2014-17 index hospitalization. Service use variables related to 30-day readmission included: one or more consultations during this period with any physician in outpatient care, psychosocial interventions in community healthcare centers, and having a usual physician. To qualify as having a usual GP, a proxy for family doctor, patients had to make at least two consultations with the same GP (in private clinics or community healthcare centers) or with at least two GPs working in the same family medicine group [33]. Some 65% of Quebec GPs work in family medicine groups that include nurses and social workers to promote higher continuity of care and extended medical coverage [34]. Having a usual psychiatrist also required at least two consultations with the same psychiatrist; for patients with at least two GP consultations, only one psychiatric consultation was necessary, which was considered a proxy for collaborative care [35]. Continuity of physician care was measured with the Usual Provider Continuity Index, describing the proportion of visits to both usual GP and psychiatrist of the total GP and psychiatrist consultations held in outpatient care settings, including walk-in clinics [36]. A score of ≥0.80 is considered high continuity of physician care [37]. Quebec community healthcare centers offer psychosocial services, notably individual or group therapy, through multidisciplinary MH teams. High ED use, defined as 3+ visits/year, reflected standard estimates [38,39]. Physician consultations measured in the 30 days following hospital discharge included those with any outpatient physician, whether GP or psychiatrist, in hospitals or the community, representing strong indicators of adequate post-discharge care [40].

### 2.3. Analysis

Descriptive analyses were conducted for the dependent variables of length of hospital stay and early readmission, and for each independent variable. Significant associations were assessed for continuous variables using ANOVA, Wilcoxon rank sum or the Kruskal–Wallis test, and for categorical variables using Chi-square tests. The intraclass correlation coefficient (ICC) was run, showing small variations among hospital centers (0.010), which precluded the need for multilevel analysis. Collinearity between variables in multivariate models was tested using variance inflation factors (VIF) and tolerance tests were run, with 5 as the maximum level of VIF. Independent variables without collinearity and with a *p*-value < 0.1 were entered into the multivariate models. Multivariate multinomial logistic regression was produced to identify predictors for length of hospitalization, with long-stay hospitalization (≥31 days) used as the reference group. Multivariable logistic regression was conducted to measure the association between length of hospital stay and 30-day readmission, controlling for sociodemographic, clinical, and service use characteristics. Odds ratios, *p*-values and 95% confidence intervals (alpha at 0.05) were calculated. The final models were selected based on the smallest Akaike’s information criterion (AIC) and Bayesian information criterion (BIC). Data analyses were conducted using STATA 17.0 software.

## 3. Results

Of the 12,000-patient cohort, 3758 were hospitalized, of whom 29 were excluded as their data on the 30-day period following discharge were not available. Of the final sample (n = 3729), 42% had brief-stay, 35% mid-stay and 23% long-stay hospitalization (Table 1), with a mean total number of inpatient days of 25.3 (SD = 45.20, median = 9, interquartile range = 25). Half (51%) were men, and 60% were 30–64 years old. Roughly 49% and 68% lived in materially and socially deprived areas, respectively. More than 57% had serious MD and 22% SRD, with a mean of 1.78 diagnosed MD/SRD (SD = 0.94), while 42% exhibited chronic physical illnesses (2.34 in mean) and 8%, suicidal behaviors. Respectively, 26% had a usual psychiatrist or both usual GP and psychiatrist, and 18% a usual GP only, with 42% receiving high continuity physician care, whereas 30% had no usual physician. Most patients (57%) received no psychosocial interventions in community healthcare centers; 18% had been hospitalized prior to their index hospitalization; and 34% were high ED users.

The rate of early readmission was 9%, although 77% received at least one consultation with an outpatient physician and 25% received psychosocial interventions from community healthcare centers within the 30-day period following discharge (Table 2).

Patients aged 65+, compared with those ≤29 years old, and patients with serious MD were more likely to have a long-stay hospitalization versus brief- or mid-stay (Table 3). Compared with long-stay patients, those with brief-stay hospitalization were more likely to have SRD or exhibit suicidal behaviors, whereas mid-stay patients had fewer chronic physical illnesses but more diagnoses of MD/SRD. Compared with long-stay patients, more brief-stay and mid-stay patients had a usual GP only, while fewer had a usual psychiatrist. Continuity of physician care was higher among brief-stay patients, who were also higher ED users than long-stay patients.

Patients with brief-stay hospitalization were more likely to be readmitted early after discharge than those with long-stay hospitalization (Table 4). Patients with early readmission also lived in more materially deprived areas, had a greater number of chronic physical illnesses and MD/SRD, and were more likely to have both a usual GP and psychiatrist or a usual psychiatrist only, compared with patients not readmitted early to hospital within 30 days.

## 4. Discussion

The mean of 25.3 days of hospitalization in this study approximates the results of a systematic review of 30 US studies reporting a mean length of hospitalization of 24.9 days [41]. Brief-stay and long-stay hospitalizations accounted respectively for 42% and 23% of total hospitalizations in our study, surprisingly high rates as both are associated in the literature with adverse outcomes [5,10]. The early readmission rate of 9% in the present study also coincided with the 8–17% range identified in previous studies [10,11]. Overall, individuals in this cohort were highly vulnerable, with most reporting serious MD as well as high material and social deprivation, nearly half had chronic physical illnesses, about a third were high ED users with no usual physician, and many did not receive any psychosocial interventions from community healthcare centers. As in previous studies [42,43], findings show that patients hospitalized for suicide attempts and MD-SRD had a higher risk of early readmission, particularly among young patients. According to a recent study [44], the risk of rehospitalization for suicidal behaviors is higher within the first month after discharge, mainly among patients with SRD, or depressive and anxious disorders. The findings confirmed the first hypothesis that patients with more complex medical conditions have a higher risk of both longer-stay hospitalization and early readmission. The fact that chronic physical illnesses often complicate treatment for patients with MD may explain the associations with long-stay hospitalization and early readmission. Serious MD also predicted long-stay hospitalization, while having a usual psychiatrist with or without a GP were the strongest predictors of early readmission: these patients were 3–4 times more likely to be readmitted early. In Quebec, primary care services exert tight control over access to psychiatric services, with long wait lists for psychiatrists who generally prioritize complex cases involving serious MD [45,46]. About two-thirds of patients with long-stay hospitalization and early readmission had a serious MD. For some of these patients, brief- and mid-stay hospitalization may therefore have been insufficient to resolve acute episodes of illness.

The number of patients with SRD and brief-stay hospitalizations in this study suggests the need to prioritize alternative treatments aimed at rapid community integration for those whose health conditions have stabilized. In Quebec, as elsewhere, SRD liaison teams have been deployed in ED and hospital units to facilitate screening, detection, and referral to community services for patients with SRD and co-occurring MD/SRD [47]. These patients can be discharged earlier and transferred to outpatient care in addiction treatment centers or groups such as Alcoholics Anonymous, who play a key supportive role for these patients [48,49]. Those with more diagnoses of MD/SRD and who tended to have mid-stay rather than long-stay hospitalization stand to benefit from the referral options mentioned above once their more complex conditions are stabilized; or, alternatively, they may be transferred to outpatient psychiatric care. Brief-stay hospitalization could also be considered for stabilized patients who exhibit suicidal behaviors, when alternative outpatient care is available. Crisis and suicide prevention services in Quebec have been greatly consolidated over the years and have developed better coordination with hospital services to facilitate follow-up care [50]. Long-stay hospitalization among patients aged 65+ is easily explained by their older age associated with higher risk of multimorbidity, which complicates treatment [51].

With few exceptions, the study findings did not confirm the second hypothesis, namely that patients receiving more intensive and continuous outpatient care would have shorter hospitalizations and fewer early readmissions. Having only a usual GP was the sole variable predicting brief-stay or mid-stay hospitalization. Studies have found that GP are uncomfortable treating patients with serious or complex MD, preferring instead to transfer them to psychiatric care [52,53]. By contrast, this study found that patients with a usual psychiatrist and high continuity of physician care—i.e., more continuous, specialized outpatient care—had more long-stay hospitalizations, perhaps due to their more complex conditions. However, long-stay hospitalization was associated with lower rates of previous high ED use among these patients, which was possibly offset by their high continuity of specialized care. Yet, having a usual psychiatrist, with or without a usual GP, strongly predicted early readmission, suggesting that the quality of this care may have been inadequate to protect these patients with high and diverse needs against early readmission.

Finally, the findings did not confirm the third hypothesis that long-stay hospitalization would increase the risk of early readmission. Instead, only brief-stay hospitalization predicted early readmission among patients who presented relatively more chronic physical illnesses or diagnoses of MD/SRD, and who lived in materially deprived areas. Brief-stay hospitalization may not have allowed clinicians to resolve complex conditions, leaving vulnerable patients more exposed to the risk of early readmission. A recent US study reported a strong association between lower socioeconomic conditions, multimorbidity, and early readmission among patients with MD [24].

It should be noted that this study has limitations. First, medical administrative databases are not primarily developed for research but for physician billing, and as such only provide proxy measures for patient health conditions. Second, key data regarding psychosocial hospital services, voluntary sector services like crisis centers, or addiction services that may have shown associations with length of hospitalization or early readmission were not available in the study database. Finally, the selected hospitals were all located in urban areas. Thus, the results may not be generalizable to semi-urban or rural territories or to other healthcare systems, particularly those without universal healthcare insurance.

## 5. Conclusions

This study found that brief-stay hospitalization (1–6 days) predicted early readmission. Patients with long-stay hospitalization (≥31 days) and early readmission also had more complex conditions, especially more co-occurring chronic physical illnesses, and more serious MD, and they tended to have a usual psychiatrist with or without a usual GP. Brief-stay hospitalization therefore needs to be promoted with care, ensuring that it has achieved patient recovery expectations. Discharge processes and the provision of outpatient care must also be adequate to prevent early readmission. As the overall cohort in this study consisted of vulnerable patients with high needs, integrating them into programs such as assertive community treatment, intensive case management or home treatment would be advisable, particularly for those with more serious MD or multimorbidity who live in materially deprived areas.

## Figures and Tables

**Figure 1 ijerph-19-15127-f001:**
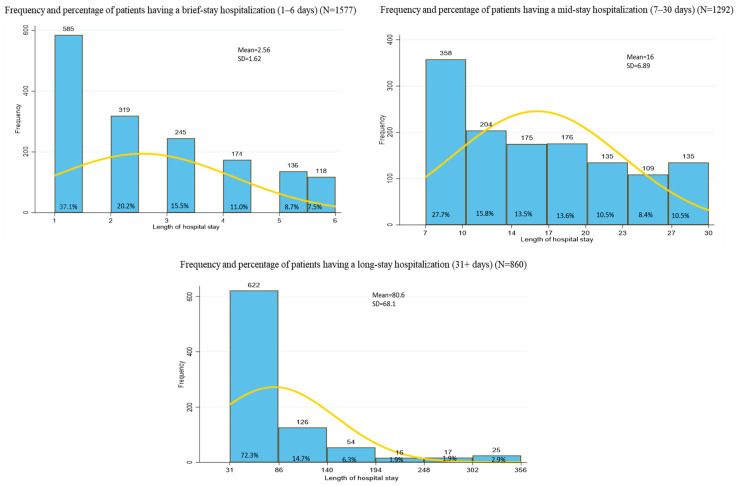
Distribution of length of hospital stay.

**Table 1 ijerph-19-15127-t001:** Characteristics of patients with mental disorders or substance-related disorders at index (first) hospitalization for mental health reasons from 2014-15 to 2016-17, according to length of hospitalization.

Variables Overall	Total Patients with Index (First Hospitalization in this 3-Year following Period) Hospitalization for Mental Health (MH) Reasons	Length of Hospitalization 2014-15 to 2016-17			Bivariate Analysis
		Brief Hospitalization: 1–6 Days	Mid-Stay Hospitalization: 7–30 Days	Long-Stay Hospitalization: ≥31 Days	
	N = 3729 (100%)	N = 1577 (42.3%)	N = 1292 (34.6 %)	N = 860 (23.1%)	
	N (%)	N (%)	N (%)	N (%)	*p* Value
**Sociodemographic characteristics** (at index hospitalization for MH reasons, 2014-15 to 2016-17)					
Sex					0.893
Men	1916 (51)	814 (52)	657 (51)	445 (52)	
Women	1813 (49)	763 (48)	635 (49)	415 (48)	
Age group					<0.001
≤29 years	1060 (28)	496 (32)	361 (28)	203 (24)	
30–64 years	2236 (60)	1002 (63)	751 (58)	483 (56)	
65+ years	433 (12)	79 (5)	180 (14)	174 (20)	
Material Deprivation Index					0.006
1–3	1896 (50.8)	774 (49.1)	703 (54.4)	419 (49)	
4–5 or not assigned ^a^	1833 (49.2)	803 (50.9)	589 (45.6)	441 (51)	
Social Deprivation Index					0.088
1–3	1198 (32)	487 (31)	445 (34)	266 (31)	
4–5 or not assigned ^a^	2531 (68)	1090 (69)	847 (66)	594 (69)	
**Clinical characteristics** (from 2012-13 to index hospitalization for MH reasons, 2014-15 to 2016-17 or other if specified)					
Mental disorders (MD) ^b^					
Common MD	2191 (59)	959 (61)	758 (59)	474 (55)	0.024
Serious MD	2117 (57)	749 (47)	776 (60)	592 (69)	<0.001
Personality disorders	658 (18)	328 (21)	203 (16)	127 (15)	<0.001
Substance-related disorder (SRD)	814 (22)	429 (27)	250 (19)	135 (16)	<0.001
Chronic physical illnesses	1565 (42)	621 (39)	545 (42)	399 (46)	0.004
Elixhauser Comorbidity Index ^c^					0.676
0	2810 (75)	1188 (75)	974 (75)	648 (75)	
1	331 (9)	148 (9)	115 (9)	68 (8)	
2	261 (7)	114 (7)	89 (7)	58 (7)	
3+	327 (9)	127 (8)	114 (9)	86 (10)	
Co-occurring MD/chronic physical illnesses	1539 (41)	609 (39)	536 (42)	394 (46)	0.003
Co-occurring SRD/chronic physical illnesses	361 (10)	192 (12)	113 (9)	56 (7)	<0.001
Co-occurring MD/SRD/chronic physical illnesses	347 (9)	184 (12)	108 (8)	55 (6)	<0.001
Number of chronic physical illnesses, mean (SD)/median (IQR)	2.34 (3.73)/ 0 (3.00)	2.12 (3.54)/ 0 (3.00)	2.29 (3.62)/ 0 (3.00)	2.82 (4.16)/ 0 (3.00)	0.004
Number of MD/SRD diagnoses, mean (SD)/median (IQR) ^d^	1.78 (0.94)/ 2.00 (1.00)	1.84 (0.99)/ 2.00 (1.00)	1.75 (0.90)/ 2.00 (1.00)	1.69 (0.88)/ 2.00 (1.00)	0.002
Suicidal behaviors (ideation or attempt related to index hospitalization, or emergency department use leading to index hospitalization)	301 (8)	196 (12)	68 (5)	37 (4)	<0.001
**Service use characteristics** (within 12 months prior to patient index hospitalization for MH reasons, 2014-15 to 2016-17)					
Usual outpatient physicians ^e^					<0.001
Usual general practitioner (GP) only	683 (18)	318 (20)	250 (19)	115 (13)	
Usual psychiatrist only	949 (26)	360 (23)	310 (24)	279 (33)	
Both usual GP and psychiatrist	963 (26)	385 (24)	325 (25)	253 (29)	
No usual physician	1134 (30)	514 (33)	407 (32)	213 (25)	
Number of consultations with usual GP ^f^					0.711
0–1 consultation	2083 (56)	874 (55)	717 (55)	492 (57)	
2 consultations	523 (14)	234 (15)	177 (14)	112 (13)	
3+ consultations	1123 (30)	469 (30)	398 (31)	256 (30)	
Number of consultations with usual psychiatrist ^g^					<0.001
0 consultation	1817 (49)	832 (53)	657 (51)	328 (38)	
1–2 consultations	394 (10)	169 (11)	134 (10)	91 (11)	
3+ consultations	1518 (41)	576 (36)	501 (39)	441 (51)	
High continuity of physician care from both usual GP and psychiatrist ((≥0.80) ^h^	1549 (42)	600 (38)	529 (41)	420 (49)	<0.001
Psychosocial interventions in community healthcare centers (excluding GP consultations)					0.006
0 intervention	2126 (57)	909 (58)	728 (56)	489 (57)	
1–2 interventions	591 (16)	268 (17)	216 (17)	107 (12)	
3+ interventions	1012 (27)	400 (25)	348 (27)	264 (31)	
Previous hospitalization for MH reasons	666 (18)	271 (17)	216 (17)	179 (21)	0.034
High emergency department use (3+ visits) for MH reasons	1251 (34)	602 (38)	410 (32)	239 (28)	<0.001

^a^ Missing address or living in an area where index assignment is not available. An index cannot usually be assigned to residents of long-term health care units or homeless individuals. ^b^ Patients may have more than one MD—total percentage may exceed 100%. Common MD included: anxiety disorders, depressive disorders, adjustment disorders, and attention deficit/hyperactivity disorder; serious MD included: schizophrenia spectrum and other psychotic disorders. ^c^ Chronic physical illnesses included: renal failure, cerebrovascular illnesses, neurological illnesses, hypothyroidism, fluid electrolyte illnesses, obesity, any tumor with or without metastasis, metastatic cancer, chronic pulmonary illnesses, diabetes complicated and uncomplicated, congestive heart failure, peripheral vascular illnesses, valvular illnesses, myocardial infarction, hypertension, pulmonary circulation illnesses, blood loss anemia, ulcer illnesses, liver illnesses (excluding alcohol-induced liver illnesses), AIDS/HIV, rheumatoid arthritis/collagen vascular illnesses, coagulopathy, weight loss, paralysis, deficiency anemia (see Appendix A). ^d^ The number of MD-SRD per patient is calculated considering the above different diagnoses: adjustment, anxiety and depressive disorders, attention deficit/hyperactivity disorder, bipolar, schizophrenia spectrum and other psychotic disorders, and personality disorders. SRD were alcohol- or drug-related disorders, use and induced disorders, intoxication, withdrawal. ^e^ Regarding usual outpatient physicians for the subgroup “both usual GP and psychiatrist”, patients must have received at least one consultation with a psychiatrist and two consultations with their GP in ambulatory care, or with two GP working in the same family medicine group (see Section 2). ^f^ Usual GP (proxy for “patient family physician”) was defined as having received at least two consultations with the same GP or with at least two GP working in the same family medicine group. These group practices require patient registration (see Section 2). ^g^ Usual psychiatrist required two consultations with the same outpatient psychiatrist, or only one outpatient psychiatric consultation plus at least two consultations with his/her GP (this was interpreted as a proxy for collaborative care). ^h^ Usual Provider Continuity Index describes the proportion of consultations with the usual GP or psychiatrist of total consultations with both physicians in outpatient care, including consultations in walk-in clinics. This index is ranked low (<0.80) or high (≥0.80). χ^2^ comparisons were provided for each row reporting percentages for categorical variables, and ANOVA or Kruskal Wallis for continuous variables.

**Table 2 ijerph-19-15127-t002:** Characteristics of patients readmitted or not for mental health reasons within 30 days after hospital discharge, 2014-17.

Variables	Total Patients at Index Hospitalization (First Hospitalization in This 3-Year following Period)	Readmission for Mental Health (MH) Reasons within 30 Days after Hospital Discharge		Bivariate Analysis
		Yes	No	
	N = 3729 (100%)	N = 321 (8.6%)	N = 3408 (91.4%)	
	N (%)	N (%)	N (%)	*p* Value
**Length of hospital stay** (main independent variable)				0.006
Brief-stay (1–6 days)	1577 (42)	160 (50)	1417 (42)	
Mid-stay (7–30 days)	1292 (35)	106 (33)	1186 (35)	
Long-stay (≥31 days)	860 (23)	55 (17)	805 (24)	
**Control variables**				
**Sociodemographic characteristics** (at index hospitalization)				
Sex				0.520
Men	1916 (51)	165 (51)	1751 (51)	
Women	1813 (49)	156 (49)	1657 (49)	
Age group				0.180
≤29 years	1060 (28)	102 (32)	958 (28)	
30–64 years	2236 (60)	190 (59)	2046 (60)	
65+ years	433 (12)	29 (9)	404 (12)	
Material Deprivation Index				0.006
1–3	1896 (51)	141 (44)	1755 (51)	
4–5 or not assigned ^a^	1833 (49)	180 (56)	1653 (49)	
Social Deprivation Index				0.043
1–3	1198 (32)	89 (28)	1109 (33)	
4–5 or not assigned ^a^	2531 (68)	232 (72)	2299 (67)	
**Clinical characteristics** (from 2012-13 [1 April–31 March] to index hospitalization 2014-15 to 2016-17, or other period if specified)				
Mental disorders (MD) ^b^				
Common MD	2191 (59)	191 (59)	2000 (59)	0.412
Serious MD	2117 (57)	213 (66)	1904 (56)	<0.001
Personality disorders	658 (18)	86 (27)	572 (17)	<0.001
Substance-related disorders (SRD)	814 (22)	88 (27)	726 (21)	0.008
Chronic physical illnesses	1565 (42)	168 (52)	1397 (41)	<0.001
Elixhauser Comorbidity Index ^c^				0.018
0	2810 (75)	220 (68)	2590 (76)	
1	331 (9)	32 (10)	299 (9)	
2	261 (7)	32 (10)	229 (7)	
3+	327 (9)	37 (12)	290 (9)	
Co-occurring MD/chronic physical illnesses	1539 (41)	168 (52)	1371 (40)	<0.001
Co-occurring SRD/chronic physical illnesses	361 (10)	44 (14)	317 (9)	0.009
Co-occurring MD/SRD/chronic physical illnesses	347 (9)	44 (14)	303 (9)	0.004
Number of chronic physical illnesses, mean (SD)/median (IQR)	2.34 (3.73)/0.00 (3.00)	3.34 (4.56)/0.00 (3.00)	2.25 (3.63)/0.00 (3.00)	<0.001
Number of MD/SRD diagnoses, mean (SD)/median (IQR) ^d^	1.78 (0.94)/2.00 (1.00)	1.77 (0.93)/2.00 (1.00)	1.85 (1.00)/2.00 (1.00)	<0.001
Suicidal behaviors (ideation, attempt related to index hospitalization, or emergency department visit leading to index hospitalization)	301 (8)	27 (8)	274 (8)	0.440
**Service use characteristics** (within 30 days of discharge from index hospitalization, or another period if specified)				
At least one consultation received with any physician in outpatient care (general practitioner (GP) or psychiatrist)	2883 (77)	305 (95)	2578 (76)	<0.001
Psychosocial interventions in community healthcare centers	939 (25)	85 (27)	854 (25)	0.308
Usual outpatient physicians (measured within 12 months prior to the 30-day period after discharge) ^e^				<0.001
GP only	438 (12)	24 (8)	414 (12)	
Usual psychiatrist only	1698 (45)	200 (62)	1498 (44)	
Both usual GP and psychiatrist	474 (13)	71 (22)	403 (12)	
No usual physician	1119 (30)	26 (8)	1093 (32)	

^a^ Missing address or living in an area where index assignment is not available. An index cannot usually be assigned to residents of long-term health care units or homeless individuals. ^b^ Patients may have more than one MD—total percentage may exceed 100%. Common MD included: anxiety disorders, depressive disorders, adjustment disorders, and attention deficit/hyperactivity disorder; serious MD included: bipolar disorders, schizophrenia spectrum and other psychotic disorders. ^c^ Chronic physical illnesses included: renal failure, cerebrovascular illnesses, neurological illnesses, hypothyroidism, fluid electrolyte illnesses, obesity, any tumor with or without metastasis, metastatic cancer, chronic pulmonary illnesses, diabetes complicated and uncomplicated, congestive heart failure, peripheral vascular illnesses, valvular illnesses, myocardial infarction, hypertension, pulmonary circulation illnesses, blood loss anemia, ulcer illnesses, liver illnesses (excluding alcohol-induced liver illnesses), AIDS/HIV, rheumatoid arthritis/collagen vascular illnesses, coagulopathy, weight loss, paralysis, deficiency anemia. ^d^ The number of MD-SRD per patient is calculated considering the above different diagnoses: adjustment, anxiety and depressive disorders, attention deficit/hyperactivity disorder, bipolar, schizophrenia spectrum and other psychotic disorders, and personality disorders. SRD were alcohol- or drug-related disorders, use and induced disorders, intoxication, withdrawal. ^e^ Regarding usual physicians for the subgroup “both usual GP and psychiatrist”, patients must have received at least one consultation with a psychiatrist and two consultations with their GP in outpatient care, or with two GP working in the same family medicine group (see Section 2). χ^2^ comparisons were provided for each row reporting percentages for categorical variables, and ANOVA and Wilcoxon rank sum for continuous variables.

**Table 3 ijerph-19-15127-t003:** Multivariate multinomial logistic regression model on index hospitalization for mental health reasons comparing brief-stay and mid-stay with long-stay hospitalization (≥31 days), 2014-15 to 2016-17 (1 April–31 March).

Variables	Brief-Stay Hospitalization: 1–6 Days				Mid-Stay Hospitalization: 7–30 Days			
	OR	*p*-Value	95% C. I		OR	*p*-Value	95% C. I	
**Sociodemographic characteristics** (at index hospitalization)								
Age (ref.: ≤29 years)								
30–64 years	0.95	0.604	0.77	1.17	0.94	0.619	0.76	1.17
65+ years	0.20	<0.001	0.14	0.29	0.62	0.003	0.45	0.85
**Clinical characteristics** (from 2012-13 to index hospitalization, 2014-15 to 2016-17, or other period if specified)								
Mental disorders (MD) ^a^								
Common MD	0.98	0.839	0.81	1.18	1.08	0.416	0.89	1.31
Serious MD	0.45	<0.001	0.37	0.55	0.77	0.013	0.63	0.94
Personality disorders	1.23	0.096	0.97	1.56	0.98	0.888	0.76	1.26
Substance-related disorder (SRD)	1.57	<0.001	1.24	1.99	1.06	0.665	0.82	1.35
Chronic physical illnesses ^b^	0.99	0.245	0.96	1.01	0.98	0.043	0.95	0.99
Number of MD/SRD diagnoses ^c^	1.03	0.079	0.99	1.06	1.05	0.001	1.02	1.09
Suicidal behaviors (ideation or attempt, related to index hospitalization, or emergency department visit leading to index hospitalization)	2.56	<0.001	1.76	3.73	1.11	0.618	0.73	1.69
**Service use characteristics** (within 12 months prior to index hospitalization, or other period if specified)								
Usual outpatient physicians (ref.: no usual physician) ^d^								
Usual general practitioner (GP) only	1.67	0.002	1.22	2.31	1.44	0.025	1.05	1.99
Usual psychiatrist only	0.68	0.008	0.52	0.90	0.61	0.001	0.46	0.80
Both usual GP and psychiatrist	0.90	0.456	0.68	1.11	0.78	0.103	0.58	1.05
High continuity of physician care from both usual GP and psychiatrist (ref. <0.80) ^e^	0.78	0.027	0.63	0.97	0.85	0.139	0.68	1.05
High emergency department use (3+ visits) for mental health (MH) reasons (ref.: <3 visits)	1.45	<0.001	1.19	1.78	1.12	0.273	0.91	1.38

^a^ Patients may have more than one MD—total percentage may exceed 100%. Common MD included: anxiety disorders, depressive disorders, adjustment disorders, and attention deficit/hyperactivity disorder; serious MD included: bipolar disorders, schizophrenia spectrum and other psychotic disorders. ^b^ Chronic physical illnesses included: renal failure, cerebrovascular illnesses, neurological illnesses, hypothyroidism, fluid electrolyte illnesses, obesity, any tumor with or without metastasis, metastatic cancer, chronic pulmonary illnesses, diabetes complicated and uncomplicated, congestive heart failure, peripheral vascular illnesses, valvular illnesses, myocardial infarction, hypertension, pulmonary circulation illnesses, blood loss anemia, ulcer illnesses, liver illnesses (excluding alcohol-induced liver illnesses), AIDS/HIV, rheumatoid arthritis/collagen vascular illnesses, coagulopathy, weight loss, paralysis, deficiency anemia. ^c^ The number of MD-SRD per patient is calculated considering the above diagnoses: adjustment, anxiety and depressive disorders, attention deficit/hyperactivity disorder, bipolar, schizophrenia spectrum and other psychotic disorders, and personality disorders. SRD were alcohol- or drug-related disorders, use and induced disorders, intoxication, withdrawal. ^d^ Regarding usual outpatient physicians for the subgroup “both usual GP and psychiatrist”, patients must have received at least one consultation with a psychiatrist and two consultations with their GP in ambulatory care, or with two GP working in the same family medicine group (see Section 2). ^e^ Usual Provider Continuity Index describes the proportion of consultations with the usual GP or psychiatrist of total consultations with both physicians in outpatient care, including consultations in walk-in clinics. This index is ranked low (<0.80) or high (≥0.80).

**Table 4 ijerph-19-15127-t004:** Logistic regression model on early readmission (within 30 days) following hospital discharge from 2014-15 to 2015-16 (1 April–31 March), controlling for patient sociodemographic, clinical, and service use characteristics (reference group: no readmission within 30 days after discharge).

	Early Readmission following Hospital Discharge			
Variables	OR	*p*-Value	95% C. I	
**Length of hospital stay** (ref.: long-stay hospitalization: ≥31 days)				
Brief-stay hospitalization (1–6 days)	1.83	<0.001	1.30	2.57
Mid-stay hospitalization (7–30 days)	1.26	0.205	0.88	1.79
**Sociodemographic characteristics** (at index hospitalization)				
Material Deprivation Index (ref.: 1–3)				
4–5 or not assigned ^a^	1.28	0.047	1.01	1.64
Social Deprivation Index (ref.: 1–3)				
4–5 or not assigned ^a^	1.12	0.416	0.85	1.47
**Clinical characteristics** (from 2012-13 to index hospitalization, 2014-15 to 2016-17, or other period if specified)				
Mental disorders (MD) ^b^				
Serious MD	1.05	0.729	0.80	1.37
Personality disorders	1.16	0.316	0.87	1.56
Substance-related disorder (SRD)	0.84	0.255	0.62	1.13
Number of chronic physical illnesses ^c^	1.058	<0.001	1.03	1.09
Number of MD/SRD diagnoses ^d^	1.18	<0.001	1.14	1.22
**Service use characteristics**				
At least one consultation received with any physician in outpatient care (general practitioner (GP) or psychiatrist) (ref.: none) within 30 days of discharge from index hospitalization, or other period if specified	1.73	0.188	0.76	3.93
Usual outpatient physicians (ref.: no usual physician) (within 12 months prior to the 30-day period after discharge) ^e^				
GP only	1.71	0.175	0.79	3.70
Usual psychiatrist only	3.88	<0.001	1.99	7.56
Both usual GP and psychiatrist	4.95	<0.001	2.46	9.98

^a^ Missing address or living in an area where index assignment is not available. An index cannot usually be assigned to residents of long-term health care units or homeless individuals. ^b^ Patients may have more than one MD—total percentage may exceed 100%. Common MD included: anxiety disorders, depressive disorders, adjustment disorders, and attention deficit/hyperactivity disorder; serious MD included: bipolar disorders, schizophrenia spectrum and other psychotic disorders. ^c^ Chronic physical illnesses included: renal failure, cerebrovascular illnesses, neurological illnesses, hypothyroidism, fluid electrolyte illnesses, obesity, any tumor with or without metastasis, metastatic cancer, chronic pulmonary illnesses, diabetes complicated and uncomplicated, congestive heart failure, peripheral vascular illnesses, valvular illnesses, myocardial infarction, hypertension, pulmonary circulation illnesses, blood loss anemia, ulcer illnesses, liver illnesses (excluding alcohol-induced liver illnesses), AIDS/HIV, rheumatoid arthritis/collagen vascular illnesses, coagulopathy, weight loss, paralysis, deficiency anemia. ^d^ The number of MD-SRD per patient is calculated considering the above diagnoses: adjustment, anxiety and depressive disorders, attention deficit/hyperactivity disorder, bipolar, schizophrenia spectrum and other psychotic disorders, and personality disorders. SRD were alcohol- or drug-related disorders, use and induced disorders, intoxication, withdrawal. ^e^ Regarding usual outpatient physicians for the subgroup “both usual GP and psychiatrist”, patients must have received at least one consultation with a psychiatrist and two consultations with their GP in outpatient care, or at least two consultations with two GP working in the same family medicine group (see Section 2).

## Data Availability

In accordance with the relevant ethics regulations for the province of Quebec, the principal investigator is responsible for preserving the confidentiality of the data.

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
