# Peer review of "Predictors of Length of Hospitalization and Impact on Early Readmission for Mental Disorders"

_ijerph, 2022, doi:10.3390/ijerph192215127_

Round 1

Reviewer 1 Report

This submission deals with an important topic in psychiatric clinical practice, that is, length of hospitalization and impact on early readmission for mental disorders. 

In all, this represents a very good piece of research. It was conducted and reported well. The Introduction provides a useful background to understand the topic. Methods and results are described and reported in detail. The Discussion is pleasant to read.

I have no major concerns, but I have two suggestions for the Authors:

1.     The study found that people with brief-stay hospitalization (1-6 days), a predictor of early readmission, were more likely to have substance-related disorders or suicidal behaviors (LL 261-262). I would suggest that the Authors add a sentence or two discussing this matter in the relevant portion of the Discussion section (LL 333-349) referring to relevant literature on this topic (Bartoli et al., 2020, https://doi.org/10.3390%2Fmedicina56110613).

2.     Figure 1 can be more cured. 

Please consider the following references as well:
  • Basith SA, Nakaska MM, Sejdiu A, Shakya A, Namdev V, Gupta S, Mathialagan K, Makani R. Substance Use Disorders (SUD) and Suicidal Behaviors in Adolescents: Insights From Cross-Sectional Inpatient Study. Cureus. 2021 Jun 11;13(6):e15602. doi: 10.7759/cureus.15602.
  • Cepeda MS, Schuemie M, Kern DM, Reps J, Canuso C. Frequency of rehospitalization after hospitalization for suicidal ideation or suicidal behavior in patients with depression. Psychiatry Res. 2020 Jan 28;285:112810. doi: 10.1016/j.psychres.2020.112810.

Author Response

Response to Reviewer 1 Comments

Point 1: The study found that people with brief-stay hospitalization (1-6 days), a predictor of early readmission, were more likely to have substance-related disorders or suicidal behaviors (LL 261-262). I would suggest that the Authors add a sentence or two discussing this matter in the relevant portion of the Discussion section (LL 333-349) referring to relevant literature on this topic (Bartoli et al., 2020, https://doi.org/10.3390%2Fmedicina56110613).

Response 1: Thank you for this suggestion. We have included a brief comment on this point in the Discussion section.

Point 2: Figure 1 can be more cured.

Response 2: Thank you for your comment. We have changed Figure 1

Point 3:  Please consider the following references as well:

Basith SA, Nakaska MM, Sejdiu A, Shakya A, Namdev V, Gupta S, Mathialagan K, Makani R. Substance Use Disorders (SUD) and Suicidal Behaviors in Adolescents: Insights From Cross-Sectional Inpatient Study. Cureus. 2021 Jun 11;13(6):e15602. doi: 10.7759/cureus.15602.

Cepeda MS, Schuemie M, Kern DM, Reps J, Canuso C. Frequency of rehospitalization after hospitalization for suicidal ideation or suicidal behavior in patients with depression. Psychiatry Res. 2020 Jan 28;285:112810. doi: 10.1016/j.psychres.2020.112810.

Response 3: The references you suggested have been added. Thank you!

Reviewer 2 Report

This study aimed to identified predictors of brief-stay (1-6 days), mid-stay (7-30 days) or long-stay (≥31 days) hospitalization and evaluated how length of hospital stay impacted  on early readmission (within 30 days) among 3,729 patients with mental disorders (MD) or substance-related disorders (SRD).

The design of the study and methodology were chosen adequately to address the objectives.

The results clearly presented. Of the 12,000-patient cohort, 3,758 were hospitalized, of whom 29 were excluded as their data on the 30-day post discharge follow-up period were not available. Of the final sample (n=3,729), 42% had a brief-stay, 35% mid-stay and 23% long-stay hospitalization (Table 1), with a mean of total number of inpatient days of 25.3 (SD=45.20, median=9, interquartile range=25). Half (51%) were men, and 60% were 30-64 years old. The findings confirmed the first hypothesis that patients with more complex medical conditions would have a higher risk of both longer-stay hospitalization and early readmission. The study findings did not confirm the second hypothesis that patients receiving more intensive and continuous outpatient care would have reduced lengths of hospitalization and early readmission. The findings did not confirm the third hypothesis that long-stay hospitalization would increase the risk of early readmission.

Conclusion. In line with the findings obtained from the study, it was determined that that brief-stay hospitalization (1-6 days) predicted early readmission. Patients with long-stay hospitalization (≥31 days) and early readmission also had more complex conditions, especially more co-occurring chronic physical illnesses, and more serious MD, while they tended to have a usual psychiatrist with or without a GP. Brief-stay hospitalization needs to be promoted with care, ensuring that it has achieved patient recovery expectations. Discharge processes and the provision of outpatient care must also be adequate to prevent early readmission. As the overall cohort in this study consisted of vulnerable patients with high needs, integrating them into programs like assertive community treatment, intensive case management or home treatment would be advisable, particularly for those with more serious MD, or multimorbidity who live in materially deprived areas

The weaker part of the study was the selected hospitals were all located in urban areas. A strong part of the study was good methodology and the design of the study.

The manuscript is suitable for printing. It is recommended to print without corrections.

Author Response

Response to Reviewer 2 Comments

Point 1: The weaker part of the study was the selected hospitals were all located in urban areas. A strong part of the study was good methodology and the design of the study.

Response 1: Thank you! This has already been pointed out in the study limitations.

Point 2: The manuscript is suitable for printing. It is recommended to print without corrections.

Response 2: Thank you very much!
